# $\mathbb{KCTS}$: Knowledge-Constrained Tree Search Decoding with Token-Level Hallucination Detection

**Sehyun Choi, Tianqing Fang, Zhaowei Wang** and **Yangqiu Song**
Department of Computer Science and Engineering, HKUST, Hong Kong SAR
{schoiaj, tfangaa, zwanggy, yqsong}@cse.ust.hk

## Abstract

Large Language Models (LLMs) have demonstrated remarkable human-level natural language generation capabilities. However, their potential to generate misinformation, often called the *hallucination* problem, poses a significant risk to their deployment. A common approach to address this issue is to retrieve relevant knowledge and fine-tune the LLM with the knowledge in its input. Unfortunately, this method incurs high training costs and may cause catastrophic forgetting for multitasking models. To overcome these limitations, we propose a knowledge-constrained decoding method called $\mathbb{KCTS}$ (Knowledge-Constrained Tree Search), which guides a frozen LM to generate text aligned with the reference knowledge at each decoding step using a knowledge classifier score and MCTS (Monte-Carlo Tree Search). To adapt the sequence-level knowledge classifier to token-level guidance, we also propose a novel token-level hallucination detection method called RIPA (**R**eward **I**nflection **P**oint **A**pproximation). Our empirical results on knowledge-grounded dialogue and abstractive summarization demonstrate the strength of $\mathbb{KCTS}$[1] as a plug-and-play, model-agnostic decoding method that can effectively reduce hallucinations in natural language generation.

## 1 Introduction

Recent progress in instruction-tuned language models (LMs) has brought forth strong general-purpose language AI that can perform well on various tasks in a zero-shot setting (Tay, 2023; Ouyang et al., 2022; Chung et al., 2022; Sanh et al., 2022). However, there have been numerous studies indicating current language models may generate non-factual information that is not supported by evidence with a high level of confidence (Ji et al., 2023a; Liu et al., 2023b). This phenomenon, often referred to as *hallucination*, poses a significant risk to the reliability of the texts generated by these models.

Previous research has attempted to mitigate this issue by augmenting the input of the language model with relevant knowledge (Asai et al., 2023), involving knowledge retrieval (Robertson and Zaragoza, 2009; Karpukhin et al., 2020; He et al., 2022; Shi et al., 2023) to identify relevant information and a reader language model that takes both the context and the retrieved knowledge as input to generate a response (Lewis et al., 2020; Izacard and Grave, 2021; Shuster et al., 2021; Borgeaud et al., 2022; Peng et al., 2023). Some works also proposed joint-train methods of the retriever and reader modules for better performance (Zhong et al., 2022b; Rubin and Berant, 2023). While these approaches have demonstrated potential, it involves pre-training or fine-tuning the reader language model, which poses significant challenges. First, the ever-increasing size of language models makes training them computationally expensive, which is becoming increasingly prohibitive, not to mention that some API-based LLMs (e.g., OpenAI APIs[2]) are not trainable by end users. Second, many state-of-the-art language models are designed to be multi-task zero-shot models through instruction tuning, aiming to perform well across various tasks. However, fine-tuning a language model extensively on a specific task can lead to catastrophic forgetting (French, 1999; Kirkpatrick et al., 2017; He et al., 2021), causing the model to lose its generalizability across different tasks and compromising its overall performance. To address these challenges, there is a pressing need for methods that do not require updating weights of language models, enabling efficient knowledge-grounded generation without sacrificing the generalization capabilities of the model.

Although designing a decoding method for LLMs is a natural way to mitigate hallucinations

---

without fine-tuning, current works in plug-and-play guided decoding (Dathathri et al., 2020; Yang and Klein, 2021; Liu et al., 2021; Chaffin et al., 2022a) are still inapt to directly be adapted to the knowledge-grounded scenarios due to their inability to identify the necessary knowledge required for generation, which leads to hallucination. Therefore, in this work, we propose a novel approach to knowledge-constrained decoding (KCD), which applies an auxiliary knowledge classifier on top of a frozen LM to detect hallucinations, and uses its knowledge-groundedness score to guide the decoding process. By incorporating these classifiers during decoding, we aim to constrain the generated text, ensuring its faithfulness to the reference knowledge. In addition, we propose a novel token-level hallucination detection method, RIPA (Reward Inflection Point Approximation), which is trained to predict the starting point of the hallucinating token and enables effective adaptation of the knowledge classifier defined on the sequence level to the token level.

[1]https://github.com/HKUST-KnowComp/Knowledge-Constrained-Decoding

To sum up, our contributions are two-fold: First, we introduce $\mathbb{KCTS}$ (Knowledge-Constrained Tree Search), a discriminator-guided decoding method that constrains the generation to be grounded on the reference knowledge, together with a novel token-level approximation method of the future reward (groundedness) through the Reward Inflection Point Approximation (RIPA). Second, our extensive experiments in knowledge-grounded dialogue and abstractive summarization tasks show the strength of $\mathbb{KCTS}$, even outperforming ChatGPT and GPT 3.5 in some dimensions.

## 2 Related Work

**Measuring Hallucinations**    Hallucination of language models or the generation of contents that are either non-factual or not supported by evidence have been studied and reported in various fields (Ji et al., 2023b; Bang et al., 2023), such as machine translation (Raunak et al., 2021), abstractive summarization (Maynez et al., 2020; Lee et al., 2022), Open Domain Dialogue (Ji et al., 2023c; Xu et al., 2023), Question Answering (Lin et al., 2022), or image captioning (Rohrbach et al., 2018). Recently developed LLMs such as Bing Chat, or perplexity.ai even serve as generative search engines, although their seemingly fluent and informative responses are not always verifiable (Liu et al., 2023b). To automatically detect and quantify hallucination

in model-generated text, several detection methods and benchmarks have been designed (Thorne et al., 2018; Pagnoni et al., 2021; Wang et al., 2020; Min et al., 2023; Manakul et al., 2023; Chen et al., 2023). In this work, instead of directly measuring hallucination, we aim to mitigate hallucination in knowledge-grounded systems, which naturally requires the response to be faithful to the knowledge.

**Knowledge Grounded Generation**    Knowledge Grounded Generation is mainly driven by retrieving relevant knowledge (Karpukhin et al., 2020; Su et al., 2022) and training the generator to produce responses augmented on the retrieved knowledge (Lewis et al., 2020; Izacard and Grave, 2021; Rashkin et al., 2021; Mialon et al., 2023). Another line of work (Févry et al., 2020; Verga et al., 2021; Zhong et al., 2022b) learns and stores entity or fact representations and provides them as input to the generator. While these methods all address the problem of knowledge-grounded generation, they all require the full fine-tuning of the generator, which may degenerate the zero-shot ability of the base model due to catastrophic forgetting, and incur a significant computational cost. A recent work (Peng et al., 2023) improves the groundedness of ChatGPT responses by incorporating the knowledge context in the prompt and providing textual feedback. While this work introduces a non-training method to knowledge-grounded generation, it is strongly dependent on the base LM's ability to understand the textual feedback and generate reference knowledge to begin with. In contrast, we propose to mitigate this problem with an approach that does not involve the fine-tuning of the generator weights and is model-agnostic.

**Guided Decoding**    Guided decoding includes supervised controllable text generation (Keskar et al., 2019; Arora et al., 2022), discriminator-guided decoding (Dathathri et al., 2020; Yang and Klein, 2021; Krause et al., 2021; Meng et al., 2022; Wang et al., 2022b) and constrained decoding (Qin et al., 2020, 2022; Lu et al., 2021, 2022; Kumar et al., 2022; Liu et al., 2023c; Geng et al., 2023), in which the user can control the sentiment or style of the generated text or constrain the generation to lexical constraints. Plug-and-Play LM (PPLM) (Dathathri et al., 2020) introduces a key concept of Bayesian decomposition $P(y|x, c) \propto P(y|x)P(c|y)$, where $c$ is the control attribute. PPLM trains a small discriminator on a frozen LM and performs gradient

ascent from the discriminator to maximize $P(c|y)$. FUDGE (Yang and Klein, 2021) instead performs weighted decoding (WD) by directly re-weighing the token probabilities $P(y_t|y_{<t}, x)$ with an auxiliary classifier probability $P(c|y_{\leq t})$. To perform re-ranking every step, $P(y|x)P(c|y)$ is decomposed into token-level and a token-level attribute classifier $P(c|y_{<t})$ is used. NADO (Meng et al., 2022) proposes to sample from a similar token-level distribution that is also weighted by $P(c|y_{<t})$, which is defined as an approximation of the sequence-level oracle $P(c|y)$. GeDi (Krause et al., 2021) and DExperts (Liu et al., 2021) also take the weighted decoding approach but avoid enumerating the vocabulary for computing $P(c|y_{<t,i})$ by training generative classifiers.

Constrained decoding methods focus on constraint satisfaction, such as lexical constraints or right-hand-side coherence (Qin et al., 2020, 2022). As constraint satisfaction can be measured after a sequence is fully generated, search-based methods (Lu et al., 2022; Chaffin et al., 2022a; Lamprier et al., 2022) that take the estimate of the future score (reward) in each decoding step has been proposed. Unlike weighted decoding, these methods commit to a token not only based on the current token's score but also on the estimate of future rewards. This may guide the generation process towards a better reward in the end.

Our method builds on guided decoding methodology but does not control a fixed set of attributes or lexical constraints but groundedness (faithfulness) to a reference knowledge.

## 3 Problem Statement

We propose to improve instruction-tuned LMs' factual generation ability under a constrained decoding setting. The problem can be formulated as

$$y \sim P_{LM}(y|x, k, \alpha_k), \quad (1)$$

where $y$ is generated text, $x$ is input text with the task description, $k$ is the piece of knowledge that $y$ must be constrained to, and $\alpha_k$ is the attribute denoting the groundedness of $y$ to $k$.

Let $f(y, k) = P(\alpha_k = 1|y, k)$ be a function that defines the groundedness of the generation $y$ to $k$. Following the Bayesian decomposition in the literature (Dathathri et al., 2020; Yang and Klein, 2021), we can apply the Bayes rule to Eq. (1) and obtain the Eq. (2) below:

$$P_{LM}(y|x, k, \alpha_k) \propto P_{LM}(y|x)f(y, k). \quad (2)$$

From an optimization perspective, obtaining a generation that is best grounded in the knowledge while being faithful to the task instruction can be written as the equation below:

$$y^* = \arg\max_y P_{LM}(y|x)f(y, k). \quad (3)$$

Then, given the auto-regressive nature of language models, Eq. (3) can be decomposed into token-level as found in FUDGE (Yang and Klein, 2021):

$$y_t^* = \arg\max_{y_t} P_{LM}(y_t|y_{<t}, x)f(y_{\leq t}, k). \quad (4)$$

**Token-Level Groundedness** Knowledge groundednss $f$ (or hallucination in the opposite perspective) is well-defined at the sequence level, which can be modeled as an entailment or fact verification problem. However, to guide the generation at each step, we need to define $f(y_{<t}, k)$ for partially generated $y_{<t}$. Following NADO (Meng et al., 2022), we define $f(y_{<t}, k)$ as the approximation of future groundedness, as denoted in Eq. (5):

$$y \sim P(y|y_{<t}, x), f(y_{<t}, k) \approx f(y, k). \quad (5)$$

## 4 The $\mathbb{KCTS}$ Method

In this section, we introduce the framework of $\mathbb{KCTS}$, which consists of a knowledge-constrained tree search decoding module (§4.1) and a token-level hallucination detection module, RIPA (§4.2). We will also introduce a sister method Knowledge Weighted Decoding (KWD) (§4.3), which is the Weighted Decoding variant using RIPA.

### 4.1 Monte-Carlo Tree Search Decoding

While weighted decoding (WD) re-weighs the token distribution with the knowledge-groundedness score at every step, it selects the most grounded token in a greedy manner, which may lead to a suboptimal solution. This is especially problematic given that the groundedness is only well-defined after the sequence is fully generated and the guidance signal from the classifier at each step is merely an approximation. To this end, we propose to use Monte-Carlo Tree Search Algorithm (MCTS) (Coulom, 2007; Chaffin et al., 2022a), which can provide a better estimate of the future knowledge groundedness through multiple simulations, as have been proven effective in other scenarios such as sentiment polarity control (Chaffin et al., 2022a,b), conditional generation (Chevelu et al., 2009; Scialom

et al., 2021; Leblond et al., 2021; Lamprier et al., 2022), and alignment (Feng et al., 2023; Liu et al., 2023a).

We will first define some notations used for the MCTS tree. Let the root node be the currently generated sequence $y_{<t}$, each node $v$ represent a token, and $\rho(v)$ be a parent node of $v$. Let $y_{<v}$ be the token sequence obtained by traversing down the tree from the root to the node $v$ and appending to $y_{<t}$ all tokens in the path except $v$.

Next, the 4 main steps of MCTS are described below in the foloowing order: Selection, Expansion, Rollout, and Backpropagation.

1. **Selection**: Starting from the root, we traverse the tree down until we reach a leaf node, selecting the children using the PUCT algorithm (Rosin, 2011; Silver et al., 2017):

$$puct(i) = \frac{V(s_i)}{n_i} + c_{puct} P(y_{s_i}|x, y_{<s_i})\frac{\sqrt{N_i}}{1+n_i}, \quad (6)$$

where $V(s_i)$ is the estimated groundedness value of node $s_i$, $n_i$ is the visit count of the node $s_i$ (i.e., number of simulations after the node), and $N_i$ is the number of visit count of the parent of $s_i$. $c_{puct}$ is a hyperparameter that controls the trade-off between exploration and exploitation, with higher $c_{puct}$ encouraging exploration. The child node $s_i$ with higher $puct(i)$ value will be selected.

2. **Expansion**: If the selected leaf node is not EOS (an end-of-sentence token, or a terminal state), the node is expanded in depth with $k$ children by decoding for one step using the LM and selecting top-$k$ tokens as the children.

3. **Rollout (Evaluation)**: from the selected leaf node $s$, generate until EOS using the language model, then evaluate the groundedness of the generated sequence, $f(y, k)$. Let this be the value of $s$, $V(s) = f(y, k)$. However, such a full rollout can be costly and result in high variance (Lamprier et al., 2022). Using the definition of $f$ in Eq. (5), the groundedness value of $f(y, k)$ can be approximated from the partially-generated sequence marked by the current node. Hence, instead of performing full rollout, we propose to directly evaluate $s$ with the approximated token-level groundedness score: $V(s) \leftarrow f(y_{<s_i}, k)$.

4. **Backpropagation**: Then, this score is back-propagated recursively from the node that

we just evaluated back to the root. Following (Chaffin et al., 2022a), we used mean aggregation of all simulations played after this node. This leads to

$$V(\rho(s_i)) \leftarrow \frac{N_i \cdot V(\rho(s_i)) + f(y_{<s_i}, k)}{n_i}, \quad (7)$$

for all $s_i$ on the path from the leaf node $s$ to the root. These values will be used in Eq. (6) to select the nodes in Step 1 in the next simulation.

These 4 steps are repeated for a predefined number of simulations, then a child node of the root that has the highest visit counts gets selected as the next token and the next root of the tree.

## 4.2 Token-Level Hallucination Detection

We first model $f$ as a fact verification problem (Thorne et al., 2018) and train a binary classifier $f(y, k) = P_f(\alpha_k = 1|y, k)$ on the sequence-level. To adapt $f$ to token-level groundedness $f(y_{<t})$, previous methods trained a classifier with random input sequence truncation (Yang and Klein, 2021; Chaffin et al., 2022b) or token-level labeling (Meng et al., 2022). The random truncation approach can be sample inefficient as only a part of the input is used during training and it may add noise to the training since the input sequence may no longer contain hallucinated content after truncation while still receiving a negative label. Although the token-level labeling approach can be more sample efficient, it may correlate benign tokens before hallucinated text with hallucination label.

**RIPA** To alleviate these shortcomings, we propose a novel approach called **R**eward **I**nflection-**P**oint **A**pproximation (**RIPA**) to approximate future $f$ for un-finished token sequence by explicitly providing a token-level label for groundedness. A schematic diagram of the comparison of RIPA and previous approaches can be found in Figure 1. Inspired by the "Hallucination Snowballing" effect (Zhang et al., 2023), where the language model's initial hallucination leads to further unsupported claims down the line, we hypothesize that identifying such an inflection point for groundedness is a more effective approximation of the future score. Hence, RIPA trains the classifier to identify the starting point, or the inflection point, of the reward (groundedness) with token-level labels that start with 1 (positive) and become 0 (negative) after the first hallucinated token. This aligns

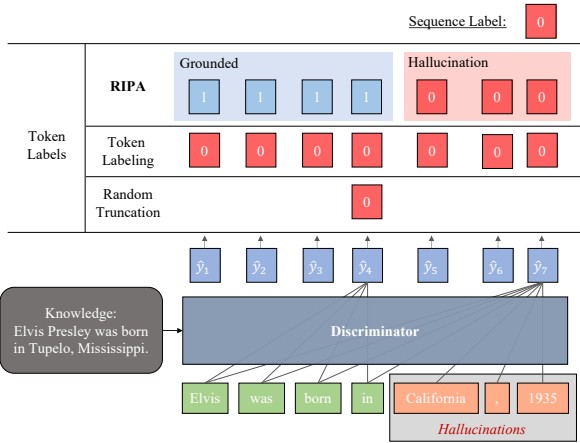

Figure 1: Our proposed token-level hallucination detection method RIPA. Previous token-level attribute classifier training randomly truncates the input sequence for classification (FUDGE) or labels all tokens in the sequence with the same sequence-level label (NADO). RIPA explicitly labels each token with groundedness, letting the discriminator associate only the hallucination sequences with the negative label.

with our definition in Eq. (5) that the label at each position $t$ to be the expected future groundedness of the preceding sequence $y_{\leq t}$: all the sequences after the first hallucination tokens include at least one hallucination token and therefore are labeled as hallucinations.

As a result, RIPA does not associate benign tokens preceding the hallucination starting point with hallucination labels, which can lead to more stable training. Additionally, it is trained to predict 0 for all tokens after hallucination is detected, which will further discount future exploration under that node in MCTS, discouraging the selection of that token. This also suggests that RIPA opens up more opportunities to explore better grounded sequences within the fixed budget of MCTS. Together, RIPA and MCTS (i.e., $\mathbb{KCTS}$) provide a better estimate of Eqs. (4) and (5), and they lead to a more efficient allocation of MCTS simulation budget and a better knowledge-constrained generation.

**RIPA Training** Training RIPA requires fine-grained token-level annotation of hallucination, which is difficult to acquire through human annotation. Alternatively, we take 2 simple approaches to generating synthetic data listed below.

1. **Knowledge Shuffle:** Given a training example $(y, x, k) \sim D$ from dataset $D$, randomly swap $k$ with another knowledge $k'$ from the knowledge source to form a negative example.

Given that the datasets contain diverse topics, $k'$ is highly likely to be irrelevant to the context $x$. Then, although the relevance between $y$ and $x$ remains unchanged, the groundedness of $y$ on $k$ becomes negative, as $y$ is no longer based on $k$. All tokens in $y$ are labeled 0.

2. **Partial Hallucination:** Similar to above, given a training example $(y, x, k) \sim D$, first randomly swap $k$ with another knowledge $k'$. Then, randomly sample a position $1 < i < T$, where $T$ is the length of $y$, and truncate the response $y$ to $i$'th token and obtain $y_i$. An LM is then asked to complete the sequence $y_i$ by sampling from $P_{LM}(y|x, y_i, k)$ in a zero-shot manner, by including the knowledge text $k$ inside the instruction. Notice that the goal here is to utilize the hallucination of LMs: hence, we sampled the completion with a temperature greater than 1. This introduces more randomness to the generation (Zhang et al., 2021; Chang et al., 2023) and allows rare tokens to be selected more easily, which in turn conditions the LM toward hallucination (Zhang et al., 2023; Aksitov et al., 2023). In this approach, only the completion tokens ($y_{>i}$) are labeled as 0.

We used a balanced mixture of the two to obtain the training set. For tasks in which $x$ and $k$ are indistinguishable (e.g., summarization), the problem becomes $P(y|k)$. Therefore, only the partial hallucination approach was employed. Detailed hyperparameters we used in each task are presented in Appendix C, and the quality analysis of the partial hallucination data is in Appendix D.

### 4.3 Knowledge Weighted Decoding (KWD)

In addition to $\mathbb{KCTS}$, we also propose a sister method named Knowledge Weighted Decoding (KWD), which applies RIPA module as the guidance classifier to the previous decoding method, FUDGE (Yang and Klein, 2021). As discussed in Related Work, FUDGE is a weighted decoding algorithm that re-ranks the top-$k$ tokens proposed by the language model using a classifier. They originally proposed to use a classifier trained with random truncation (§4.2, also see Fig. 1), which might be inoptimal for knowledge grounded generaiton. On the other hand, KWD uses RIPA as the guidance classifier module for improved guidance signal at the token-level. This method serves as a

bridge between previous methods and $\mathbb{KCTS}$ and will be used in the ablation studies.

## 5 Experiments Setup

In this section, we will first describe the tasks selected for knowledge-constrained decoding (§5.1), then the evaluation metrics used (§5.2). Finally, the baselines and implementation details are provided in §5.3 and §5.4.

### 5.1 Datasets

To show the strength of the guided decoding method in the knowledge-grounded generation, we have selected two well-studied tasks from the literature: knowledge-grounded dialogue and abstractive summarization. In both tasks, the language model is given a piece of reference knowledge in the input and asked to generate a response using that knowledge.

**Knowledge Grounded Dialogue**  Knowledge-grounded dialogue (KGD) can be formulated as modeling $P_{LM}(y|x, k)$, where $y$ is the generated response, $x$ is dialog history, and $k$ is the relevant knowledge. We experimented with gold knowledge, as the focus of this study was to show the potential of constrained decoding in knowledge grounding. We used the Wizard of Wikipedia (WoW) dataset's (Dinan et al., 2019) unseen topic portion of the test set as the benchmark dataset for this task.

**Summarization**  Abstractive summarization can be naturally considered as a knowledge-grounded generation task, as the generated summary should only contain the information from the reference document. In fact, improving factual consistency in abstractive summarization is a challenging task (Pagnoni et al., 2021; Wang et al., 2020). We used CNN/DM (See et al., 2017) dataset as the benchmark dataset of our study.

### 5.2 Evaluation Metrics

We use various evaluation metrics applied for knowledge grounding and natural language generation. We categorize the metrics into 3 categories: token-based, knowledge-based, and multifaceted. For token-based automatic metrics, we used BLEU-4 (Papineni et al., 2002), Rouge-L (Lin, 2004), ChrF (Popović, 2015), and ME-TEOR (Banerjee and Lavie, 2005), following Peng et al. (2023). For knowledge-based metrics, we

first used Knowledge-F1 (KF1; Lian et al., 2019), which measures the unigram overlap between the generated and knowledge tokens, and K-Copy, as defined in Eq. (8),

$$1 - \frac{LD(y, k)}{\max(|y|, |k|)}, \tag{8}$$

where $LD$ stands for Levenshtein Distance between generated response and reference knowledge string. This metric captures the amount of verbatim copies of the knowledge in a generation. The purpose of this metric is to monitor if the model simply copies the knowledge as a response, which defeats the purpose of using a generative LM. Hence, an excessively high copy rate (e.g., $\geq 70\%$) may indicate a reduced utility of the response.

Finally, we also utilize UniEval (Zhong et al., 2022a), a multifaceted, model-based evaluator trained using Boolean QA format. For the dialog task, we utilize Naturalness, Coherence (with dialogue context), and Groundedness (to the knowledge), and for summarization, we take Coherence (within the summary), Consistency (with the article), Fluency, Relevance (to the gold answer) as fine-grained evaluation dimensions. For summarization, we also employ MFMA (Lee et al., 2022) pre-trained metric, which showed SOTA-level correlation with human labels on CNN/DM (See et al., 2017) data split of FRANK (Pagnoni et al., 2021) and QAGS (Wang et al., 2020) benchmark.

### 5.3 Baselines

We use popular API-based LLMs and publicly available instruction-tuned language models of various sizes as the initial baseline. We experimented with the LLMs provided through OpenAI API; namely, ChatGPT (`gpt-3.5-turbo-0301`) and GPT 3.5 (`text-davinci-003`). For instruction-tuned models, we studied two different sizes (XL & XXL) of the Flan-T5 (FT5) model family (Chung et al., 2022), and T0++ (Sanh et al., 2022). Note that they are not directly compared with our method due to the significant differences in terms of the model size and training cost.

While FT5 and T0++ models have been fine-tuned on some dialogue data, knowledge-grounded dialogue is still an unseen task for these models. Hence, we first gathered zero-shot results from various instruction-tuned models and experimented with guided decoding. On the other hand, CNN/DM summarization task is included in the T0 dataset mixture and the Natural Instructions

| Type | Model | K-Overlap | | Token Overlap | | | | | UniEval | | | $f$ |
|---|---|---|---|---|---|---|---|---|---|---|---|---|
| | | KF1 | K-Copy | F1 | BLEU | RougeL | ChrF | METEOR | N | C | G | |
| LLM | ChatGPT | 49.41 | 39.71 | 30.32 | 6.91 | 26.24 | 34.95 | 31.67 | 57.62 | 96.41 | 96.15 | 95.82 |
| | GPT-3.5 | 25.91 | 28.22 | 22.32 | 3.01 | 18.70 | 27.86 | 23.06 | 42.77 | 98.07 | 92.42 | 92.63 |
| SFT | FT5-XL | 39.85 | 37.79 | 28.08 | 9.41 | 25.11 | 31.17 | 25.40 | 76.44 | 92.36 | 95.16 | 97.90 |
| Zero-Shot | FT5-XL | 34.50 | 37.07 | 21.18 | 6.81 | 19.64 | 24.88 | 18.53 | 71.69 | 82.21 | 75.70 | 88.75 |
| | FT5-XXL | 28.20 | 32.33 | 19.11 | 5.53 | 17.55 | 24.15 | 17.16 | 72.37 | 84.24 | 75.51 | 85.89 |
| | T0++ | 26.94 | 28.80 | 17.57 | 4.13 | 16.14 | 19.84 | 13.37 | 52.79 | 85.26 | 70.14 | 88.61 |
| Decoding Baselines | FUDGE | 55.30 | 54.04 | 29.43 | _11.72_ | 27.35 | 31.50 | 26.00 | 73.68 | 88.20 | 83.53 | 94.54 |
| | NADO | 50.20 | 50.10 | 27.86 | 10.57 | 26.01 | 29.84 | 24.51 | _74.14_ | 88.35 | 81.10 | 92.76 |
| | MCTS | 55.54 | 54.21 | 29.56 | 11.69 | _27.48_ | 31.60 | 26.08 | **74.54** | 88.16 | 83.90 | 95.07 |
| Ours | KWD | **58.19** | 56.58 | **30.71** | **12.74** | **28.27** | _33.40_ | _28.10_ | 70.27 | _90.51_ | _87.86_ | _97.54_ |
| | КCTS | _56.06_ | 51.90 | _30.54_ | 11.42 | 27.43 | **35.22** | **28.92** | 62.32 | **92.78** | **91.78** | **98.30** |

Table 1: Results on WoW Test set (unseen topics). SFT stands for supervised fine-tuning, and FT5 is shorthand for Flan-T5. Under the UniEval metrics, each letter stands for the following: **N** - Naturalness, **C** - Coherence, **G** - Groundedness. For all metrics, a larger number is preferred, except for K-Copy. Note that the performance of LLM in the upper half is for reference only. For each column, **boldface** denotes the best score out of the KCD methods under the FT5-XL backbone, and underline indicates the second best.

| $T$ | K-Overlap | | Token Overlap | | UniEval | | $f$ |
|---|---|---|---|---|---|---|---|
| | KF1 | K-Copy | BLEU | RougeL | C | G | |
| 5 | 48.78 | 48.22 | 10.17 | 25.39 | 90.58 | 85.87 | 90.58 |
| 10 | 48.24 | 48.05 | 9.98 | 25.87 | 90.22 | 86.41 | 85.43 |
| 16 | 51.49 | 48.67 | 11.07 | 26.44 | 92.83 | 89.99 | 92.76 |
| 32 | 56.06 | 51.90 | 11.42 | 27.43 | 92.78 | 91.78 | 98.30 |

Table 2: Ablation study on the number of initial tokens to be constrained in the knowledge with КCTS.

dataset (Wang et al., 2022a), which was part of the Flan finetuning (Chung et al., 2022). Therefore, performing Knowledge-Constrained Decoding (KCD) on CNN/DM test set can be considered as guiding an already-finetuned model to improve the factuality dimension further.

Then, we apply weighted decoding (WD) & constrained decoding baselines, namely FUDGE (Yang and Klein, 2021), NADO (Meng et al., 2022), and MCTS (Chaffin et al., 2022a), on the KCD setting directly, which serve as the strong baselines directly comparable to our method.

### 5.4 Implementation Details

To train the classifiers, we applied lightweight adapters through LoRA (Hu et al., 2021) only to the decoder layers of the language model and added a single linear layer on top of the last hidden states. This only adds 0.21% of additional training weights that can be disabled or enabled at test time, which does not hurt the rich multi-task ability of the base instruction-following model. See Appendix C for more details about model training.

## 6 Main Evaluation

In this section, we provide the evaluation results on the above-mentioned two tasks and conduct more analysis to demonstrate the reasons behind the success of our method.

### 6.1 KGD Results

**Results Analysis** We report the performance of zero-shot LLMs and various instruction-finetuned models in the upper half of Table 1 and the performance of directly comparable decoding-based baselines and our methods in the lower half. We also studied the performance of a Supervised-FineTuned (SFT) version of FT5-XL for the KGD task. Note that the performance on the upper half (LLM and SFT) is only used to provide an overviewed understanding of how powerful each language models are when tested on WoW and are *not* directly compared to our methods. The instructions used for each model are listed in Appendix A.

From the results in the upper half of Table 1, ChatGPT shows a strong ability to generate responses that show high overlap with the knowledge. On the other hand, FT5-XL, while being the smallest in size (3B) out of all the models studied, showed the best performance in generating responses that are the most grounded to the reference knowledge, as indicated by the KF1 and Groundeness column of the Table 1. Therefore, we selected FT5-XL as our base model in our further studies for guided decoding[3].

---

[3]We also conducted a preliminary experiment of KCD application to GPT-3.5 in Appendix B.

| Type | Model | K-Overlap | | Token Overlap | | | | | UniEval | | | | MFMA |
|---|---|---|---|---|---|---|---|---|---|---|---|---|---|
| | | KF1 | K-Copy | F1 | BLEU | RougeL | ChrF | METEOR | Coh. | Cons. | fluency | Relv. | score |
| **LLM** | ChatGPT | 29.43 | 17.92 | 40.45 | 11.75 | 27.85 | 42.96 | 37.66 | 93.85 | 91.67 | 87.15 | 87.11 | 80.62 |
| | GPT-3.5 | 27.54 | 16.94 | 38.96 | 10.78 | 26.63 | 41.17 | 35.38 | 92.56 | 90.33 | 85.73 | 85.78 | 78.74 |
| **SFT** | FT5-XL | 17.04 | 10.18 | 32.21 | 8.74 | 24.02 | 30.27 | 24.47 | 84.82 | 86.02 | 89.90 | 81.28 | 64.55 |
| | FT5-XXL | 17.45 | 10.42 | 31.55 | 8.43 | 23.38 | 29.95 | 23.91 | 87.17 | 88.58 | 90.00 | 82.28 | 68.37 |
| | T0++ | 22.79 | 13.65 | 38.82 | 13.64 | 28.06 | 38.53 | 33.68 | 86.57 | 87.47 | 89.03 | 81.09 | 69.38 |
| **Decoding Baselines** | FUDGE | 18.68 | 10.70 | 33.51 | 9.32 | 24.83 | 31.06 | 24.93 | 90.52 | 90.61 | 83.37 | 82.00 | 71.35 |
| | NADO | 20.35 | 11.72 | 35.10 | 10.93 | 26.22 | 33.50 | 27.34 | 92.26 | 93.72 | 88.41 | 84.49 | 72.01 |
| | MCTS | 17.86 | 10.04 | 34.59 | 9.00 | 25.85 | 30.90 | 25.12 | 94.30 | 94.28 | 86.51 | 85.90 | 71.28 |
| **Ours** | KWD | 20.39 | 11.63 | 36.24 | 12.30 | 27.20 | 34.25 | 28.46 | **96.24** | **96.64** | **91.60** | **88.48** | 85.11 |
| | 𝕂ℂ𝕋𝕊 | **22.97** | 13.29 | **38.27** | **14.21** | **28.10** | **37.18** | **31.37** | 95.85 | 96.03 | 90.24 | 87.16 | **85.36** |

Table 3: Results on CNN/DM Test set. The guided decoding was conducted with FT5-XL model as the base model. **Coh.**, **Cons.**, and **Relv.** stand for coherence, consistency, and relevance, respectively. As the performance of LLMs is for reference, we highlight the best scores on the last two groups with **boldface** and second-best with underline.

The comparison of baseline decoding methods and our proposed 𝕂ℂ𝕋𝕊 can be found in the lower half of Table 1. The penultimate group contains the results of baseline decoding methods, guided by their own proposed approximation of token-level classifiers. FUDGE and MCTS both use random truncation approximation, and NADO uses a token-level labeling approach. All decoding methods showed improvement over the nucleus sampling baseline regarding groundedness, indicated by a higher KF1 score and Groundedness column of UniEval. The results also clearly show that the RIPA provides a better token-level guidance signal for KCD, as both KWD and 𝕂ℂ𝕋𝕊 outperform baselines in all dimensions except the naturalness. 𝕂ℂ𝕋𝕊 also resulted in the highest $f$ activation, confirming the hypothesis that MCTS, which also estimates future rewards for token selection through simulations, can produce a higher reward.

**Does 𝕂ℂ𝕋𝕊 really estimate future groundedness?** To show that 𝕂ℂ𝕋𝕊 guides the future generation trajectory towards a more grounded response in the end, we experimented with constraining the token generation for initial $T$ tokens, then letting the original language model complete the sentence with nucleus sampling. The results in Table 2 indicate that the initially grounded tokens provide a good context to the LM that leads to a more grounded response generation, following the intuition of using MCTS with RIPA to fulfill the definition of future groundedness $f(y_{<t}, k) \approx f(y \sim P(y|y_{<t}), k)$. Moreover, this provides an additional performance/speed trade-off parameter.

| | Model | Fl. | Relv. | Gr. | K-Copy |
|---|---|---|---|---|---|
| Overall | ChatGPT | 3.00 | 2.73 | 2.62 | 0.04 |
| | FT5-XL | 2.64 | 2.30 | 1.95 | **0.12** |
| | FUDGE | 2.82 | 2.35 | 2.19 | 0.21 |
| | 𝕂ℂ𝕋𝕊 | **2.92** | **2.55** | **2.37** | 0.17 |
| Non -Copy | ChatGPT | 3.00 | 2.75 | 2.60 | - |
| | FT5-XL | 2.61 | 2.31 | 1.81 | - |
| | FUDGE | 2.78 | 2.40 | 1.97 | - |
| | 𝕂ℂ𝕋𝕊 | **2.91** | **2.61** | **2.24** | - |

Table 4: Human Evaluation in 3-point Likert scale on WoW Test Set. **Fl.**, **Relv.**, and **Gr.** stands for Fluency, Relevance, and Groundedness, respectively. The inter-rater agreement by Krippendorff alpha (Krippendorff, 2011) was 0.57, 0.46, 0.77, 0.31. Non-Copy means average scores of examples that annotators agreed the generation does not copy the knowledge.

**Human Evaluation** We also conducted a human evaluation of the generated responses to assess their quality. We randomly sampled 100 examples and asked 3 different evaluators to measure their fluency, relevance to the dialogue context, groundedness to reference knowledge, and if the response is an unnatural copy of the knowledge. The human evaluation results in Table 4 further confirm the strength of 𝕂ℂ𝕋𝕊, which received better scores in all dimensions than the baselines. Furthermore, 𝕂ℂ𝕋𝕊 resulted in higher relevance and groundedness while copying less knowledge than FUDGE, which suggests that 𝕂ℂ𝕋𝕊 responses have higher perceived utility. The results in the Non-Copy group also show that 𝕂ℂ𝕋𝕊 outperforms baselines even excluding the knowledge-copied responses.

| Model | Fluency | Grounded | Complete |
|-------|---------|----------|----------|
| ChatGPT | 3.00 | 2.93 | 2.88 |
| FT5-XL | 2.81 | 2.60 | 2.13 |
| FUDGE | 2.89 | 2.90 | 2.31 |
| KCTS | **2.95** | **2.97** | **2.40** |

Table 5: Human Evaluation on CNN/DM. This follows a 3-point Likert scale, with agreement alpha of 0.35, 0.44, and 0.19.

## 6.2 Summarization Results

**Results Analysis** We used the same models as in §6.1. From the results found in Table 3, it can be observed that ChatGPT again outperforms other models in most dimensions, except for BLEU and Rouge metrics. On the other hand, the instruction-tuned models show a different trend than with the KGD setting; T0++ model outperforms FT5 models, presumably because Flan finetuning introduces a vast amount of additional tasks than T0 finetuning dataset mixture, leading to a deteriorated performance in some of the tasks. This finding aligns with our motivation for not finetuning the base LM directly for knowledge grounding.

For efficiency, we have continued to use the FT5-XL model throughout guided decoding experiments with the summarization task. KCTS again showed superior performance over the baseline methods, with significant improvements in token overlap and MFMA. RIPA-guided decoding also outperformed all baseline methods in UniEval metrics, with KWD showing a slightly better performance than KCTS. This might be partially attributed to KCTS having higher knowledge overlap with the original article (KF1 = 22.97) than KWD (KF1 = 20.39), which may suggest the summaries were "less abstractive" and situated in the out-of-distribution of the training data of the UniEval model. Overall, knowledge-based, reference-based, and learned metrics consistently support the effectiveness of KCTS.

**Human Evaluation** We have randomly selected 50 samples for human evaluation, with the same 3 human evaluators from the KGD task. The evaluators were asked to evaluate the summaries in 3 dimensions: fluency, groundedness, and completeness. Groundedness is analogous to the precision of knowledge, while completeness is similar to recall. It can be observed from Table 5 that the evaluators preferred KCD over the baselines in all dimensions, including groundedness and completeness.

## 7 Conclusion

In this work, we proposed KCTS, a new guided decoding approach to knowledge-grounded natural language generation. Our token-level hallucination detection module RIPA used with MCTS decoding has shown effectiveness in knowledge-grounded dialogue and abstractive summarization tasks regarding both automatic metrics and human evaluation, surpassing previous guided decoding baselines and even ChatGPT in some metrics. Applying KCTS only on the first few tokens also proved helpful in the knowledge-grounded generation, adding another dimension of performance/speed trade-off.

## Limitations

One limitation of the approach is additional computational cost and inference time. For NADO, which computes $f(y_{<t,i}, k)$ for all $i \in \mathcal{V}$ once, it requires two forward passes (1 for LM, 1 for discriminator) per token. FUDGE needs to enumerate the top-$k$ tokens to feed into the discriminator, taking $k$ discriminator forward passes per token. For KCTS (MCTS), it takes $N$ LM and discriminator forward passes each to perform $N$ simulations per token. However, we assert that the generation speed and groundedness are in a tradeoff relationship, as the $k$ or $N$ can be reduced for a faster generation. In future work, the time complexity can be further reduced by training a smaller auxiliary weight for discriminator or employing early-stopping heuristics to reduce the number of simulations of MCTS decoding (Baier and Winands, 2012).

As this study focused on showing the constrained-decoding methods' strengths in the knowledge-grounded generation, we did not consider knowledge retrieval and experimented with gold knowledge. Constraining on retrieved knowledge can be considered to test in a more realistic deployment scenario.

ChatGPT was generally preferred over smaller instruction-tuned models during human evaluation, even with KCD applied. However, since the details about training data, model architecture, or mechanisms behind ChatGPT are not public, we cannot ascertain that this is a fair comparison. Moreover, as our method is model-agnostic, it could also be applied to large language models by those with access to further improve them.

## Ethics Statement

Our experiments are conducted on two datasets, namely the Wizard of Wikipedia (WoW) dataset (Dinan et al., 2019) for Knowledge Grounded Dialogue and CNN/DM (See et al., 2017) dataset for abstractive summarization. The knowledge part in WoW is retrieved from Wikipedia, which is open-access, and the project[4] is under MIT license without any copyright issues. CNN/DM is a well-studied summarization dataset that is crawled from CNN and Daily Mail news, where the data[5] is under an apache-2.0 license that is safe to use for research studies. Both datasets do not involve privacy problems about any specific entities (e.g., a person or company) and are widely used in NLP research papers.

Human evaluation was performed by 3 postgraduate NLP researchers with at least 1 year of experience in the field to ensure quality. Two of the three annotators employed for the human evaluation were authors of the paper, and the third annotator was another postgraduate student in NLP within the same institution for a broader and more objective perspective. The authors' involvement in the annotation process was part of their academic responsibilities, and no additional compensation was provided. The third annotator was compensated for their time and effort at the hourly rate equivalent to 7.65 USD/hr, which was in line with the university guidelines and higher than the local law of minimum wage.

Our method focuses on factual natural language generation in terms of groundedness to the reference knowledge. It does not verify the factual correctness of the provided reference, so if the knowledge source contains non-factual information, KCD is likely to also generate misinformation. We urge the users to verify the knowledge source carefully before usage.

## Acknowledgements

The authors of this paper were supported by the NSFC Fund (U20B2053) from the NSFC of China, the RIF (R6020-19 and R6021-20), and the GRF (16211520 and 16205322) from RGC of Hong Kong. We also thank the support the UGC Research Matching Grants (RMGS20EG01-D, RMGS20CR11, RMGS20CR12, RMGS20EG19, RMGS20EG21, RMGS23CR05, RMGS23EG08).

---

[4] https://github.com/facebookresearch/ParlAI
[5] https://huggingface.co/datasets/cnn_dailymail

We would also like to thank the Turing AI Computing Cloud (TACC) (Xu et al., 2021) and HKUST iSING Lab for providing us computation resources on their platform.

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

## A Instruction Templates

The instruction used for different models and tasks are listed in Table 6. For ChatGPT with KGD task, we used the chat completion API, where each dialogue turn is separated and formatted as a user/assistant message, and the instruction was given as the system message at the end.

## B Application to LLMs

As a preliminary study, we have experimented with applying knowledge-constrained decoding on LLMs, GPT-3.5 (text-davinci-003) in our study. One limitation of the OpenAI API is that it does not return the token probability distribution over the whole vocabulary; at most top-5 log probabilities are returned. This significantly limits the search space for all WD methods, which may reduce the ability to guide the generation toward the objective. Hence, we propose a new method called *Pre-KWD*[6], where we use a proxy model to propose top-$k$ tokens first, re-rank the tokens with RIPA, then include it in the API request in the *logit bias* field, which is added to the logit of the LLM before sampling. This can be denoted as:

$$Z_i = Z_i^{LLM} + \alpha \Big[ \tilde{Z}_i + \log f(y_{<t,i}, k) \Big] \quad (9)$$

Where $Z_i$ is the logit of a token $i$, $Z^{LLM}$ is the logit of the base LLM, and $\tilde{Z}$ is the logit of the smaller proxy model. $\alpha$ is another hyperparameter that controls the strength of logit bias. Since GPT2 (Radford et al., 2019) shares its vocabulary with GPT3 family, $\tilde{Z}$ was computed with GPT2-XL.

We have randomly sampled 100 examples from the WoW test set for this experiment. The results in Table 7 shows that applying post-guidance does not improve much, as the search space is limited: without sufficient width, the generation is bound to what the base model believes. On the other hand, although the overlap between tokens proposed by the proxy model and actual token distribution is unknown, the empirical results suggest

---

[6]In turn, we call the original KWD as *Post*-KWD here.

| Model | Task | Instruction |
|---|---|---|
| FT5, T0, GPT3.5 | Summarization | ### Document:
ARTICLE

Given the article, generate a faithful summary. |
| | KGD | History:
DIALOG

Knowledge:
KNOWLEDGE

Given the dialog history and a relevant knowledge,
generate a knowledgeable, useful, and helpful response. |
| ChatGPT | Summarization | Summarize the following text:
ARTICLE |
| | KGD | {content: turn 1, role: user}
{content: turn 2, role: assistant}
...
{content: Use the following knowledge, but not directly copy,
to generate a concise response: "KNOWLEDGE",
role: system} |

Table 6: Instruction templates for different models for different tasks.

| Decoding | K-Overlap | | Token Overlap | | | | | UniEval | | |
|---|---|---|---|---|---|---|---|---|---|---|
| | KF1 | K-Copy | F1 | BLEU | RougeL | ChrF | METEOR | N | C | G |
| GPT-3.5 | 25.75 | 28.40 | 23.71 | 3.91 | **20.20** | 28.53 | **24.46** | 40.42 | 98.70 | 94.19 |
| *Post*-KWD | 26.94 | 29.53 | 23.80 | 3.41 | 19.78 | 28.58 | 24.32 | **45.62** | 97.80 | 94.12 |
| *Pre*-KWD | 27.44 | 29.15 | 23.86 | 3.91 | 19.93 | 28.02 | 23.64 | 39.24 | **98.90** | 94.43 |
| *Pre+Post*-KWD | **27.92** | **30.51** | **24.00** | **4.00** | 19.96 | **28.83** | 23.97 | 41.43 | 98.72 | **95.18** |

Table 7: GPT3.5 + KWD on 100 random examples from WoW test set (unseen topics).

that this method can successfully add bias toward tokens that are grounded on reference knowledge. Finally, using both pre-guidance and regular post-reweighting together can result in the most faithful generation.

**Limitations** Applying KWD to LLM incurs $O(T)$ times more cost, where $T$ is the number of generated tokens. This is because we need to query the API 1 token at a time, leading to redundant computation of the prompt tokens. This can be easily mitigated with attention key-value caching by the API provider, which we hope to be enabled in the future.

## C Implementation Detail

The response length was set to 64 tokens for summarization and 32 for KGD. For nucleus sampling, top-$p$ was 0.95 with temperature = 1, which also applies to OpenAI models. For all decoding methods studied, we applied top-$k$ filtering with $k = 50$. In addition, in NADO, the constraining factor $\alpha$ was set to 0.25, and in MCTS, the constant $c_{puct} = 3$ and the number of simulations $N = 50$. We also

applied repetition penalty (Keskar et al., 2019) of 1.2 for MCTS following the original implementation.

The synthetic data generated has the following statistics: for WoW, 8,832 partial hallucination examples were generated using FT5-XL in a zero-shot manner, with temperature $T = 1.4$ to encourage hallucination. 10,000 knowledge shuffle examples were also sampled, along with 20,000 original examples, leading to a balanced mixture of 20k positive examples and 18.8k negative examples. For CNN/DM, 12,811 partial hallucination negative examples were generated using the same procedure. The final dataset included 13,180 positive examples as well. We then applied a random 9:1 split to obtain the training and test sets.

All experiments were conducted with NVIDIA GPUs with CUDA 11.7 with CuDNN 7.5$\geq$ enabled. We used either RTX A6000 48GB or RTX 3090 24GB GPUs. All classifier training was performed with an effective batch size of 64 for KGD and 32 for summarization for 2000 steps. For efficiency, we loaded the models in 8-bit quantization with

bitsandbytes (Dettmers et al., 2022a,b), both during training and inference.

All implementations were based on the huggingface transformers (Wolf et al., 2020) and peft (Mangrulkar et al., 2022) library. We also utilized evaluate[7] library for metric implementations. All the pretrained model weights were downloaded from huggingface hub[8].

## D Analysis on Partial Hallucination Data

In this section, we evaluated the utility of the generated Partial Hallucination data in two aspects: hallucination success and relevance to practical decoding scenarios.

To evaluate the hallucination success rate, we compared the groundedness score of the original to the partial hallucination data through UniEval. The score drop was from 0.95 to 0.63 (KGD groundedness), and from 0.88 to 0.36 (Summarization Consistency), which suggests that our silver-standard partial hallucination label is, to a large extent, valid.

Another concern is that the high-temperature sampling used to generate partial hallucination data may lead to sequences that the model does not normally generate. Then, the discriminators trained on this dataset may not be exposed to the usual tokens generated by the LM and diminish their utility in guided decoding. To ascertain the relevance of the hallucinated sequences, we examined if the sampled tokens with high temperature fall into the top-50 most probable tokens of our base LM (flan-t5-xl), as this was the search width for each step of KWD and KCTS. It was found that 68% of the sampled tokens for summarization (78% for Knowledge Grounded Dialogue) are included in the top-50, which suggests that the synthetic negative data generated using high temperature are relevant to the real use-case.

## E Classifier Performance

We report the performance of the knowledge-groundedness classifiers used for decoding on the test split of the pseudo-dataset generated in Figure 2. Notice that both dialogue and summarization datasets are balanced (Appendix C), so comparing accuracy would suffice. We also included the accuracy of the sequence-level groundedness classifier $f$ for reference.

[7]https://huggingface.co/docs/evaluate/index
[8]https://huggingface.co/models

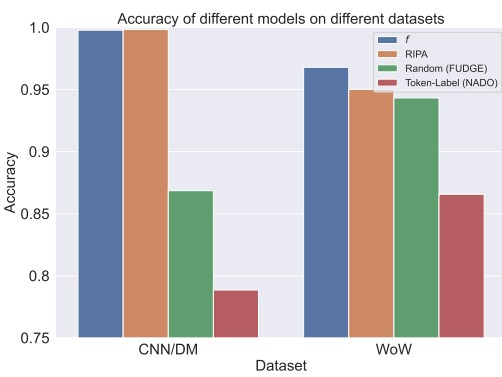

Figure 2: Classifier performance in test split of pseudo-data generated for CNN/DM and WoW.

From the results in Figure 2, we can first observe that $f$ achieves very high accuracy, suggesting that its a good validator of groundedness defined by the dataset. RIPA closely follows $f$, outperforming other token-level classifiers. The random truncation approach shows suboptimal performance, which aligns with our hypothesis that this approach may add noise during training. Token-labeling approach from NADO shows the lowest accuracy, which can be attributed to their classifier design not being standard binary classifier; their classifier outputs $V$ independent output scores $Z_i$ such that $Z_i = f(y_{<t,y_i}, k)$, where $V$ is vocab size, and $y_i$ is the $i$-th token in the vocabulary. This can be interpreted as having $V$ different independent classifiers. While this allows for faster inference, some of the classifier may receive far less number of training examples depending on the token distribution in the training dataset. Moreover, since this approach labels all tokens with the same label, it also suffers from the problem of associating benign tokens before hallucination with negative label. RIPA, on the other hand, alleviates this problem and consequently achieves the best accuracy out of the token-level classifiers.

## F Generated Examples

We included some example outputs from each method for the case study for KGD in Table 8 and summarization in Table 9. Example 1&2 of Table 8 show the strength of our approach. In Example 1, while the FT5-XL fails to address the knowledge entirely and FUDGE outputs "sweater" which is not mentioned, $\mathbb{KCTS}$ uses the knowledge in a natural way to generate the response. ChatGPT also generates a well-grounded response. In example 2,

| | | |
|---|---|---|
| **Example 1** | **Context** | - hey, do you like cardigans? |
| | **Knowledge** | a more modern version of the garment has no buttons and hangs open by design. |
| | ChatGPT | Yes, I do like cardigans. I particularly like the more modern version that has no buttons and hangs open by design. |
| | FT5-XL | I love cardigans. |
| | FUDGE | A cardigan is a sweater with buttons. |
| | 𝕂ℂ𝕋𝕊 | I like cardigans, but i don't like the buttoned version. |
| **Example 2** | **Context** | - i love dogs! my favorite breed is the goldendoodle! they're a crossbreed between and golden retriever and a poodle.
- i have seen one before and they are so cute.
- they are so fluffy and cute right?! the original purpose of the cross was actually to develop guide dogs suitable for those with allergies to dogs!
- has it been successful for that purpose? |
| | **Knowledge** | their coats do not shed, which reduces dander. |
| | ChatGPT | Yes, the goldendoodle's coat doesn't shed much, making them great for people who suffer from allergies. The breed's hypoallergenic coat means... |
| | FT5-XL | That's true! |
| | FUDGE | Yes, they are very popular with people with allergies. |
| | 𝕂ℂ𝕋𝕊 | Yes, they are a great breed for people with allergies because their coats do not shed which reduces the amount of dander that can be... |
| **Example 3** | **Context** | - i drink protein powder with nothing but water .
- really? have you ever tried fish protein powder? sounds gross, but its made for humans to drink.
- does it smells and taste good. maybe i could give it a trial. i am trying to gain muscles here. |
| | **Knowledge** | fish protein powders have various sanitary processing, purity and functional characteristics which establish them as human food ingredients. |
| | ChatGPT | Fish protein powder is actually a popular supplement among bodybuilders due to its high protein content and amino acid profile. It is made through a sanitary processing method to ensure... |
| | FT5-XL | yes it smells very good and tastes very good. |
| | FUDGE | yes, it is a good source of protein. |
| | 𝕂ℂ𝕋𝕊 | yes, it is a good source of protein and it has a good taste. |

Table 8: Example generations selected from WoW test set.

FT5-XL is not very informative, FUDGE makes an unsupported claim about goldendoodle's popularity, while 𝕂ℂ𝕋𝕊 generates the most knowledgeable response. On the other hand, Example 3 shows the failure case for all models. While the provided knowledge is about "fish protein powders" being sanitary, ChatGPT mentions high protein content and other models mention taste.

Table 9 includes two success cases for KCD. ChatGPT response tends to be more detailed, which may contribute to having higher completeness in human evaluation, but it can be not sufficiently concise to serve as a good summary. FT5-XL may generate seemingly unrelated comments, and FUDGE can degenerate in fluency during token re-ranking. 𝕂ℂ𝕋𝕊 generates the most concise and faithful summary that captures the core of the article.

| | | |
|---|---|---|
| **Example 1** | **Article** | (CNN)Their relationship led to jail time for her, but Mary Kay Letourneau Fualaau wants the world to know that she and her husband, Vili Fualaau, are still happily together. She was a married 34-year-old teacher and mother of four in Seattle in 1996 when she began an affair with Fualaau, her 13-year-old student. Letourneau gave birth to her young lover's child and went on to serve more than seven years in prison on charges related to their sexual relationship. On Friday, ABC's "20/20" will air an exclusive interview with the couple, who will celebrate their 10th wedding anniversary on May 20. The pair wed soon after she was released from prison in 2005 and are now the parents of two teen girls. According to ABC, during the interview with Barbara Walters, "Mary Kay tells Walters what makes their marriage work in spite of their huge difference in their age." "She also discusses her surprising plans to teach again, as well as her intentions to have her status as a registered sex offender lifted," a posting on the network's site says. "Vili Fualaau, meanwhile, discusses his bouts with alcoholism, depression and why he believes the system failed him while he was still a minor." Letourneau Fualaau is now 53, and her husband is 31. |
| | ChatGPT | Mary Kay Letourneau Fualaau and her husband, Vili Fualaau, are still happily together and are celebrating their 10th wedding anniversary on May 20. Letourneau Fualaau served over seven years in jail for having an affair with Fualaau, her 13-year-old student, |
| | FT5-XL | It's been more than 10 years, and no, that's not a big difference in age. |
| | FUDGE | A married teacher and mother of four in Seattle in 1996 began an affair with her student, Vili Fualaau, who was a teen then a teen-ager. |
| | KCTS | Mary Kay Letourneau Fualaau, who gave birth to her young lover's child and went on to serve more than seven years in prison, will tell Barbara Walters how she and her husband, Vili Fualaau, are still happily together. |
| **Example 2** | **Article** | Tim Sherwood's penchant for a gilet has put the Aston Villa boss' sense of fashion in the spotlight and it isn't hard to imagine what he thought about Carles N'Zogbia's choice of attire on Tuesday. The Villa wide man turned up for training at the club's Bodymoor Heath complex wearing an outrageously flamboyant shirt and trouser combo. Shay Given certainly wasn't a fan of N'Zogbia's flowery ensemble, made worse by his decision to wear sunglasses in the dressing room. Charles N'Zogbia poses in his flowery outfit before Aston Villa training on Tuesday . N'Zogbia has featured regularly under Tim Sherwood since his appointment in February . N'Zogbia is no stranger to a garish outfit as seen here in a camouflage jump suit . Tim Sherwood became famous for wearing a gilet during his time as Tottenham manager last season . The Aston Villa goalkeeper took to Twitter to ridicule the Frenchman's choice of attire describing it as the 'the worst ever'. The Ireland international posted a picture of N'Zogbia on the social networking site writing 'OMG get on Charles N'Zogbia's gear. #wow #Worstever'. N'Zogbia has featured regularly under Sherwood since he was appointed in February, but after catching a glimpse of his outfit may have had a few choice words for the former Newcastle man. Villa moved six points clear of the Premier League relegation zone after Saturday's 1-0 win against Tottenham at White Hart Lane and face Liverpool in the FA Cup semi-final at Wembley next weekend. |
| | ChatGPT | Aston Villa footballer Carles N'Zogbia caused a stir when he turned up to training wearing an outrageous floral shirt and trousers combo, topped off with sunglasses in the dressing room. Even goalkeeper Shay Given ridiculed the outfit on Twitter, calling it "the worst ever". |
| | FT5-XL | Villa keeper Shay Given trolled Carles N'Zogbia's flowery outfit. |
| | FUDGE | Aston Villa's Carles N'Zogbia wore a garish outfit to training on Tuesday and Tim Sherwood slammed 'the worst ever' goalkeeper Shay Given. |
| | KCTS | Aston Villa's Carles N'Zogbia turned up for training on Tuesday wearing a flowery shirt and trouser combo. Shay Given took to Twitter to ridicule the Frenchman's choice of attire describing it as the 'worst ever'. |

Table 9: Example generations selected from CNN/DM test set.