# OpenReview forum: "KCTS: Knowledge-Constrained Tree Search Decoding with Token-Level Hallucination Detection"
_EMNLP/2023/Conference — EMNLP 2023 Main_

### Official Review · Reviewer_jKoz · 2023-08-02

**Soundness:** 4

**Ethical Concerns:**

Yes

**Excitement:**

4: Strong: This paper deepens the understanding of some phenomenon or lowers the barriers to an existing research direction.

**Justification For Ethical Concerns:**

There is no mention about how evaluators were rewarded by their labour.

**Paper Topic And Main Contributions:**

The paper describes a knowledge-constrained tree search decoding mechanism applied on top of a large language model to reduce hallucinations. The paper is well written, clear and the related work is well described. Authors propose to guide decoding through a Monte-Carlo tree search algorithm which estimates the future groundedness. The Reward Inflection Point Approximation (RIPA), a binary classifier, is then used to predict the inflection point where a token is not longer grounded to the knowledge source. For training RIPA sequences were automatically generated by knowledge shuffle or by response truncation plus generation with high temperature. This mechanism was evaluated on two datasets the wizard of Wikipedia and CNN/DM for grounded dialogue and summarisation respectively. Distinct metrics were used such as token-based (BLEU, ROUGE, METEOR), knowledge-based (Knowledge-F1, K-Copy) and UniEval. For summarisation the metric MFMA was also used to find correlations with human labels. The results show the effectiveness of the proposed method KCTS. My unique concern regards the technique of Knowledge-Weighted Decoded (KWD), or the RIPA guided variant, which is not explained in the paper.

**Questions For The Authors:**

Why Knowledge-Weighted Decoded (KWD), the RIPA guided variant, is not clearly explained? How does it perform an ablation study?

**Reasons To Accept:**

The paper describes a knowledge-constrained tree search decoding mechanism applied on top of a large language model to reduce hallucinations. The paper is well written, clear, detailing the related work. Authors propose to guide decoding through a Monte-Carlo tree search algorithm which estimates the future groundedness. The Reward Inflection Point Approximation (RIPA), a binary classifier, is then used to predict the inflection point where a token is not longer grounded to the knowledge source. For training RIPA sequences were automatically generated by knowledge shuffle or by response truncation + generation with high temperature. This mechanism was evaluated on two datasets the wizard of Wikipedia and CNN/DM for grounded dialogue and summarisation respectively. Distinct metrics were used such as token-based (BLEU, ROUGE, METEOR), knowledge-based (Knowledge-F1, K-Copy) and UniEval. For summarisation the metric MFMA was also used to find correlations with human labels. The results show the effectiveness of the proposed methods KCTS.

**Reasons To Reject:**

The Knowledge-Weighted Decoded (KWD), the RIPA guided variant which is not clearly explained in the paper.

**Reproducibility:**

3: Could reproduce the results with some difficulty. The settings of parameters are underspecified or subjectively determined; the training/evaluation data are not widely available.

**Reviewer Confidence:**

4: Quite sure. I tried to check the important points carefully. It's unlikely, though conceivable, that I missed something that should affect my ratings.

---

> ### Author Rebuttal · Authors · 2023-08-29
>
> We appreciate your constructive feedback. The following responds to the questions, which simultaneously addresses the reasons to reject.
>
> **Details of KWD**
>
> The reason we did not explain KWD in more detail was because KWD is a method that combines RIPA and weighted decoding (WD), which was explained thoroughly in the related work (line 174). KWD can be considered as a RIPA classifier-guided FUDGE.
>
> **How does it relate to ablation studies**
>
> KWD stands in the middle of Weighted Decoding (WD) baselines (FUDGE, NADO*) and KCTS, because KWD adds RIPA classifier on WD, and KCTS further uses MCTS Decoding with RIPA. Therefore, FUDGE/NADO vs KWD shows the effectiveness of RIPA by keeping the decoding algorithm the same, and KWD vs KCTS shows the effectiveness of RIPA + MCTS.
>
> **Although FUDGE and NADO are both categorized as WD, they have a different implementation of the classifiers: FUDGE employs a binary classifier and enumerates over top-k tokens to perform WD, while NADO’s classifier outputs a $V$-dimensional vector ($V$ is the vocabulary size) that can be added to the logits directly. This is also explained in lines 630-632 of the paper, and further detail can be found in NADO paper (Meng et al., 2022). KWD uses the approach in FUDGE.*
>
> **References**
>
> Tao Meng, Sidi Lu, Nanyun Peng, and Kai-Wei Chang. 2022. Controllable Text Generation with Neurally-Decomposed Oracle. In Advances in Neural Information Processing Systems, volume 35, pages 28125–28139. Curran Associates, Inc.

---

### Official Review · Reviewer_nrcv · 2023-08-04

**Soundness:** 4

**Excitement:**

4: Strong: This paper deepens the understanding of some phenomenon or lowers the barriers to an existing research direction.

**Paper Topic And Main Contributions:**

This paper proposes a decoding method to reduce hallucinations in text generation.
They apply an auxiliary knowledge classifier on top of a frozen LM to detect hallucinations, and use its knowledge-groundedness score to guide the decoding process.
In addition, they propose a token-level hallucination detection method to predict the starting point of the hallucinating token and enables effective adaptation of the knowledge classifier to the token level.

**Reasons To Accept:**

1. This paper proposes a novel token-level hallucination identification method and a knowledge-constrained decoding method to reduce hallucinations in instruction-tuned LLMs.
2. This paper conducts extensive experiments and evaluations to validate the effectiveness of the proposed decoding method.

**Reasons To Reject:**

N/A

**Reproducibility:**

3: Could reproduce the results with some difficulty. The settings of parameters are underspecified or subjectively determined; the training/evaluation data are not widely available.

**Reviewer Confidence:**

3: Pretty sure, but there's a chance I missed something. Although I have a good feel for this area in general, I did not carefully check the paper's details, e.g., the math, experimental design, or novelty.

---

> ### Author Rebuttal · Authors · 2023-08-29
>
> We sincerely appreciate the time and effort invested by the reviewer in thoroughly evaluating our paper.

---

### Official Review · Reviewer_rGKp · 2023-08-04

**Soundness:** 4

**Excitement:**

3: Ambivalent: It has merits (e.g., it reports state-of-the-art results, the idea is nice), but there are key weaknesses (e.g., it describes incremental work), and it can significantly benefit from another round of revision. However, I won't object to accepting it if my co-reviewers champion it.

**Paper Topic And Main Contributions:**

This paper proposes a Knowledge-Constrained Tree Search decoding method that is more faithful to the reference knowledge provided to the model. Together with this method, there is a token-level approximation method to predict if the generated content is grounded in knowledge (Reward Inflection Point Approximation, RIPA). In addition, there are experiments in knowledge-grounded dialogue and abstractive summarization tasks where the paper method outperforms GPT models in some dimensions.

One problem that the authors tackle is to understand at token-level when the output starts to be not grounded in the knowledge. This is well-defined at sentence level, i.e. after the full generation, when a classifier can be used to get a desired signal. At token level, they use a Monte-Carlo Tree Search Algorithm to explore multiple generation paths, and instead of using a groundedness signal from a classifier at the end of a generation only, they incorporate the same signal for partial paths.

The authors describe RIPA, a model that detects the first hallucinated token in the sequence generation, trained on synthetic data obtained from knowledge swapping/shuffling to create grounded and ungrounded (random) instances.

Automatic metrics and human evals show that the models using RIPA are better than the baselines.

**Questions For The Authors:**

Eq (5): going from string to probability in the function f would require some more explanation or more specific pointers to the NADO paper sections.

In the Rollout paragraph of section 4.1, what is the criteria to stop when generating partial sequences, if EOS is not used since it is too costly?

I am not sure about the strong statement at line 344 that RIPA does not associate benign tokens with hallucination labels, since facts that appear after an hallucinated token may still be grounded (e.g., Barack Obama is Norwegian and his wife is Michelle Obama).

When training RIPA, how can you guarantee that the examples obtained with the Partial Hallucination method have the last part that is actually based on the provided knowledge? What if the model does not use the knowledge and outputs something from its own knowledge? I imagine that using high temperature to create more random sequences is yielding examples which are very different from what would be sampled during decoding with the normal temperature.

**Reasons To Accept:**

Reducing hallucination and improving groundedness is an hot and important topic in the LLM research community and any effort in this direction is extremely valuable and of interest.

The method is slightly better (larger for summarization) than the baselines on the automatic metrics but is significantly better according to the human evaluations.

**Reasons To Reject:**

The synthetic data generation methods used to train RIPA may have some issues since it seems that there are no guarantees that generated tokens are hallucinated or not factual.

According to human evals, KWD is better for summarization than the proposed method. KWD still uses RIPA but it is only described with a quick one liner. Human evals, which are more important, may tell the more nuanced story that KCTS may not be always the best RIPA guided method, and require a bit of restructuring of the paper.

**Reproducibility:**

3: Could reproduce the results with some difficulty. The settings of parameters are underspecified or subjectively determined; the training/evaluation data are not widely available.

**Reviewer Confidence:**

4: Quite sure. I tried to check the important points carefully. It's unlikely, though conceivable, that I missed something that should affect my ratings.

---

> ### Author Rebuttal · Authors · 2023-08-29
>
> We appreciate your detailed feedback. The following paragraphs respond to the reasons to reject together with the question.
>
> **Additional explanations to RIPA**
>
> *1. Guarantee of hallucination in partial-hallucination data*
>
> The intuitive guarantees lie in two aspects. First, as the knowledge is randomly sampled, and both of our datasets contain diverse topics, it’s highly likely the random knowledge is irrelevant to the context. In this sense, if the LM uses the knowledge provided, it’s expected to generate ungrounded outputs, i.e., hallucination. Second, high temperature sampling in decoding will bring larger randomness, leading to more irrelevant content. This phenomenon has been reported in the NLG literature (Zhang et al., 2020, Chang et al., 2023) where larger temperature brings randomness and diversity while sacrificing quality. While the LM may contain parametric knowledge about the context, high temperature allows more rare tokens to be easily selected, which in turn conditions the LM toward hallucination, as pointed out in Zhang et al. (2023) and Aksitov et al. (2023).
>
> It is a valid concern that high temperature sampling may lead to sequences that the model does not normally generate. We consider this to be a trade-off between hallucination success rate and quality. Moreover, we checked if the sampled tokens with high temperature fall into the top-50 most probable tokens of our base LM (flan-t5-xl), as this was the search width for each step of KWD and KCTS. It was found that 68% of the sampled tokens for summarization (78% for Knowledge Grounded Dialogue) are included in the top-50, which suggests that the synthetic data generated using high temperature are relevant to the real use-case.
>
> In addition, when we observed the groundedness score difference from the original to the partial-hallucination data through UniEval, the score drop was 0.95 -> 0.63 (KGD groundedness), and 0.88 -> 0.36 (Summarization Consistency), which suggests that our silver-standard partial hallucination label is, to a large extent, valid.
>
>
> *2. RIPA and benign tokens*
>
> The intention behind the statement was that RIPA does not associate benign tokens “before” the hallucination with the hallucination label, which NADO or FUDGE may do. Given the autoregressive nature of LM and the definition of $f$ having to approximate the groundedness of the whole sequence that would follow the current sequence, we believe the labels should become 0 for all tokens after hallucination. In other words, if the partial-sequence only consists of grounded tokens, the full sequence following it has a chance to be grounded, but if it contains at least one hallucinated content, all generations that can follow from it will be considered hallucinated sequences.
>
> Practically, this assumption helps MCTS decoding to reject hallucination token better. As RIPA is trained to predict 0 for all tokens following hallucinated tokens, all descendants of a hallucination token in the MCTS tree will be evaluated to have lower scores, which will in turn decrease the value of the hallucination token through backpropagation. This will prevent the token from being selected.
>
> **KWD showing better results in human eval**
>
> We respectfully ask for your clarification on “human eval”, since KWD was not included in the human evaluation. In the following paragraphs, we will assume that you are referring to the unieval results on the summarization task (CNN/DM dataset, Table 3 in our paper), in which KWD achieved a higher score than KCTS.
>
> We would first like to stress that UniEval is a model-based learned metric, which is strongly influenced by its training data. We hypothesize that higher knowledge overlap (KF1 of KCTS=22.97 vs KWD=20.39) with the original article might indicate the generated summary was slightly “less abstractive”, which might situate in the out of distribution of the summarizations included in the training of UniEval metrics. Also, we would like to point out that other dimensions of evaluation in summarization tasks, including Knowledge Overlap, Token Overlap, and learned (MFMA), prefer KCTS consistently. Overall, along with the results on the Knowledge-Grounded Dialogue (KGD) task, most knowledge based, reference based, and learned metrics show the effectiveness KCTS.
>
> In addition, the $f$ column of Table 1 shows that MCTS decoding leads to generations that the groundedness classifier finds more grounded (looking at rows FUDGE/NADO vs MCTS and KWD vs KCTS). This finding supports our hypothesis that MCTS decoding guides the generation towards higher reward of $f$ at the end of sequence. We believe this is another evidence that KCTS is the better-suited RIPA-guided decoding method.
>
> We will include more details about KWD and the discussion above in the final version of our paper.
>
> **Probability in Eq. (5)**
>
> Thank you for pointing out the misnotation of Eq. (5); the input of function $f$ should not have been probability but the decoded token sequence. We will make the update in the final version as follows:
>
> $$
> y \sim P(y|y_{<t}, x), f(y_{<t}, k) \approx f(y, k).
> $$
>
> **Stopping criteria for the Rollout**
>
> To avoid expensive rollout, we evaluated the node directly; that is, we did not generate any tokens following the node but applied $f$ right on top of the node. Generative Cooperative Network (Lamprier et al., 2022) is a previous work that also employed MCTS with NLG, and they had the same approach of not performing rollout to avoid high variance (which we have missed the citation at line 288; we will also update this in the final version). We further formulated the discriminator score of partial sequence ($y_{< t}$) as the approximation of full sequence following it ($y$) as in Eq. (5), which would make the prediction more stable.
>
>
> **References**
>
> Hugh Zhang, Daniel Duckworth, Daphne Ippolito, & Arvind Neelakantan. (2020). Trading Off Diversity and Quality in Natural Language Generation.
>
> Chung-Ching Chang, David Reitter, Renat Aksitov, & Yun-Hsuan Sung. (2023). KL-Divergence Guided Temperature Sampling.
>
> Muru Zhang, Ofir Press, William Merrill, Alisa Liu, and Noah A. Smith. 2023. How language model hallucinations can snowball.
>
> Renat Aksitov, Chung-Ching Chang, David Reitter, Siamak Shakeri, & Yunhsuan Sung. (2023). Characterizing Attribution and Fluency Tradeoffs for Retrieval-Augmented Large Language Models.
>
> Sylvain Lamprier, Thomas Scialom, Antoine Chaffin, Vincent Claveau, Ewa Kijak, Jacopo Staiano, and Benjamin Piwowarski. 2022. Generative cooperative networks for natural language generation. In Proceedings of the 39th International Conference on Machine Learning, volume 162 of Proceedings of Machine Learning Research, pages 11891–11905. PMLR.

---

### Meta-Review · Area_Chair_baWZ · 2023-09-16

**Recommendation:** 4

**Metareview:**

KCTS, a knowledge-constrained decoding method, guides a frozen LM to generate text aligned with the reference knowledge at each decoding step using a knowledge classifier score and MCTS. KCTS is a plug-and-play, model-agnostic decoding method that can effectively reduce hallucinations in natural language generation without incurring high training costs or causing catastrophic forgetting.
The proposed approach is shown to be effective in both automatic and human evaluations.

---

### Decision · Program_Chairs · 2023-10-07

**Decision:**

Accept-Main

**Comment:**

KCTS, a knowledge-constrained decoding method, guides a frozen LM to generate text aligned with the reference knowledge at each decoding step using a knowledge classifier score and MCTS. KCTS is a plug-and-play, model-agnostic decoding method that can effectively reduce hallucinations in natural language generation without incurring high training costs or causing catastrophic forgetting.
The proposed approach is shown to be effective in both automatic and human evaluations.